# Investigating Possible Effects from a One-Year Coach-Education Program

**DOI:** 10.3390/sports9010003

**Published:** 2020-12-26

**Authors:** Frode Moen, Maja Olsen, John Anders Bjørkøy

**Affiliations:** 1Department of Education and Lifelong Learning, Faculty of Social and Educational Sciences, Norwegian University of Science and Technology, 7491 Trondheim, Norway; 2Centre for Elite Sports Research, Department of Neuromedicine and Movement Science, Faculty of Medicine and Health Science, Norwegian University of Science and Technology, 7491 Trondheim, Norway; olsenmajagunhild@gmail.com; 3The Norwegian Centre for Elite Sports, Sognsveien 228, 0840 Oslo, Norway; JohnAnders.Bjorkoy@olympiatoppen.no

**Keywords:** coach–athlete relationship, coach education, working alliance, mentoring

## Abstract

The main purpose of the current study was to examine possible effects from a coach education program over one year, in which each coach was supervised by a mentor who facilitated their learning based on coach-centered values. The current study was designed as an experiment with a control group, where the coaches in the experiment group received mentoring by a mentor over one year, whereas the coaches in the control group did not. Ninety-four coaches completed the study over one year from a variety of sports (*n* > 30), where cross-country skiing, soccer, biathlon, handball and swimming were the most represented sports. Among the coaches in the sample, 87% were coaches for athletes who competed or had ambitions to compete at an international level. The results from self-reported questionnaires at the pre-test and post-test show that the coach education program had a significant effect on the bond dimension in the coach–athlete working alliances and the coaches’ perceived coach performances. The analysis did not find any effects from the coach education program on the goal and task dimension in the coach–athlete working alliances. The findings are discussed in terms of applied implications and possible future research.

## 1. Introduction

One essential part of high-performance sport is the competitive nature where athletes’ capabilities are tested against one or more opponents. In this competitive environment, athletes are the key performers who need to maintain or develop their skills and capacities over time to be competitive. However, in the process of developing an athlete, coaches play an important role [1]. The fact that increased financial investments in coaching staffs are used to explain successes in elite sports supports the argument that coaches are key for athletic success [2,3,4,5]. An important part of such increased investments involves educating and strengthening the competence of the coaching staff. Thus, coaches are key for athlete development and it is not surprising that there are numerous scientific studies within sports claiming that a well-functioning coach–athlete relationship is essential in developing athletes and teams [1,6,7,8,9,10,11].

Importantly, the primary output of coaches’ education in high-performance sport should eventually lead to athlete development and enhanced performances [12]. The key responsibility for a coach in high-performance sport is therefore to ensure athletes’ growth and learning and increasing their competitiveness. As a consequence, the education of coaches has been “a very hot topic” for several years [13] (p. 145) and several countries have invested substantial finances to optimize and redesign their coach education [14]. The substantial increase in coach education delivery in many Western nations [13,15,16], is based on the claim that the importance of coach education “cannot be overestimated” to raise the quality among coaches [17] (p. 275). It is not surprising then that a great deal of resources are used to find and hire the best coaches, to formally educate upcoming coaches, and to continually send coaches to different seminars and courses to ensure their professional development in their role. Educating and developing coaches to develop their competencies is essential in all sports. Unfortunately, research claims that coaches rarely find formal academic coaching education programs important or useful for their roles [15,18,19,20]. Based on research that claims the impact from coach education programs is limited [21,22,23,24,25,26,27,28] and the fact that the financial investments in coach education have increased, possible effects from such programs should therefore continue to be scientifically investigated.

## 2. Theory

It is claimed that formal courses in coach education are governed too much by prescriptive and rigid rationales [15,18,19,20]. Formal sources of learning are defined as education programs that are institutionalized and fulfil official academic quality demands [29]. On the other hand, research claims that both elite and non-elite coaches mainly use non-formal and informal sources in their learning process [21,30,31,32]. Non-formal sources are organized outside the formal academic system and are not bound to any quality demands from an official curriculum [29]. As a consequence, coaches seek informal sources of learning that positively influence their coaching practice and knowledge on their own. Informal sources of learning include reflections upon own experience as an athlete or a coach, discussions with other coaches, books, and information found on the internet, that ultimately develop their competence [29,33]. Thus, it is mainly their own experience, the sharing of experience with other coaches and the observation of other coaches that remain the primary sources of knowledge acquisition for coaches according to research [15,19]. Therefore, to succeed with education programs for coaches, their desire to become better practitioners through pedagogical approaches that actively involve them and their experiences seem to be essential [34].

Recent research claims that effective coach education should consider the obvious power and influence from experience in coaches’ learning, and the importance of learning from other coaches and their experience [15]. Thus, coach education needs to include supervised field experiences in a variety of contexts and enable the coaches to reflect on their experiences and learn from them. The optimal coach education from coaches’ subjective perspectives seem to be a combination of non-formal and informal sources of knowledge in their learning process, where reflections are facilitated by another person based upon the coaches’ own experiences. Interestingly, recent research suggests that mentoring is a potential effective sport coaching development tool [35]. Mentoring is the process of receiving guidance and support by a more experienced person who serves as a mentor, and mentoring can be executed both informally and formally [36]. Informal mentoring is spontaneous, not managed, structured or formally recognized, whereas formal mentoring is sanctioned, managed and structured by an organization [35].

Research claims that both sport practitioners and scholars need to expand their thinking about coaching and coach education and consider if learning processes should be based on a facilitative approach or a teaching approach [35,37]. A coach-centered approach focuses on the potential and empowerment of the coach and is a typical facilitative approach to learning [38,39]. When a coach-centered approach is used in coach education it has the potential to strengthen the coaches’ feeling of being autonomous because the power is given to them, not “lent” until it no longer benefits the educator that traditionally is more powerful [37]. When power is given to the coaches in the learning process, they will be aware that the strategies they develop in order to reach their goals derive from their own experience and competence [40,41,42,43]. This is found to positively stimulate both coaches’ and athletes’ intrinsic motivation and self-confidence [44,45,46].

**The coach–athlete relationship.** Recent research indicates that the creation of strong bonds, clear goals and effective affiliated strategies related to these goals, are important parts that describe the effectiveness of the coach–athlete relationship [43,47]. These components, constituting the working alliance between coaches and their athletes [48], require that there is be a mutual agreement between coaches and their athletes concerning what goals they are trying to achieve, as well as what tasks these goals demand in order for both parties to succeed [47,49]. Importantly, the strategy dimension constitutes athletes’ experience of improvement towards agreed upon goals. Additionally, the collaboration between coaches and their athletes require strong relational bonds, referring to the level of liking, caring and trusting established between coaches and their athletes [47,50,51]. Originally, the working alliance was first discussed as a concept between therapists and clients in psychotherapy; however, “a working alliance between a person seeking change and a change agent can occur in many places besides the locale of psychotherapy” [48] (p. 252). Hence, this can also occur in the sport context. In fact, it is of special interest here because it considers goal achievement and perceived performance development in addition to the empathic bonds [47]. Thus, both relational and performance aspects are considered. Due to the competitive nature of sports, this seems paramount [47]. A successful coach–athlete working alliance based on these criteria will most likely enhance athletes’ performances and their sport competitiveness, and thus, the coaches will be more effective in their roles. Thus, based on the theoretical arguments presented above an effective coach education program should have a significant impact on coaches’ perceptions of the coach–athlete working alliance and their perceptions of their performances as coaches.

**The present study.** The main aim of the current study is to investigate possible effects of a one-year coach education program based on a formal mentoring program, considering the following hypotheses:

**Hypothesis** **1.**
*The coach education program will improve coaches’ perceptions of their coach–athlete working alliances with their athletes through the dimensions bond, goal and task.*


**Hypothesis** **2.***The coach education program will improve coaches’ perceptions of their performances as coaches*.

## 3. Materials and Methods

The Norwegian Olympic Sports Center (NOSC), the national organization that is part of the Norwegian Olympic and Paralympic Committee and Confederation of Sports, initiated promising coaches to a two-year coach education project where the goal was to develop coaches of young and promising athletes. 

**Participants** were recruited from the Norwegian education coach program arranged by the NOSC. The NOSC has responsibility for training and management of elite coaches and athletes. Coaches in all parts of Norway were openly invited to apply for the program. To apply and be selected, the coaches had to be prioritized by their sport federations and preferably be 30 years or younger. A total of 185 coaches applied for the program from all over Norway. In all, 109 coaches were selected to participate in the program. Out of the 109 coaches, 107 accepted the invitation to participate in the study. From the 107 coaches that participated at the pre-test, 94 completed the data collection after 1 year, which gives a response rate of 88%. The sample consisted of 61 males (65%) and 33 females (35%) whose ages ranged from 23 to 44 years (*M* = 29.8 *SD* = 3.83). The participants practiced a variety of sports (*n* > 30) including both team and individual sports, although nearly half of the sample practiced either cross-country skiing (18.1%), soccer (11.7%), biathlon (10.6%), handball (5.3%) or swimming (7.4%). Among the coaches in this sample, 87.1% worked as coaches for athletes who competed at an international level or had ambitions to compete at an international level, while 12% worked with athletes from recreational sports. Seventy seven percent of the coaches were the head coach for their teams, whereas 23% were assistant coaches. From the sample of coaches, 45.7% were fully employed as a coach, 44.7% were employed part-time and 9.6% worked voluntarily as a coach. Eighty four percent of the coaches had education at university level (Bachelor 55.3%, Master 27.7%, PhD 1.1%) while 16% had no education after high school. The coaches worked 27 h a week as coaches on average (2 h minimum and 80 h maximum per week). They had an average of 8 years of experience as coaches (1 year min and 18 years max).

**Pre-test/post-test control group design.** The current study was arranged as an experiment with a control group. The 107 coaches were assigned into the experimental or the control group based on their applications. The program was arranged to educate approximately 50% of the coaches the first year (from January 2019), while the other 50% would serve as a control group the first year (2019). In the second year, the roles were exchanged (from January 2020). The applications had one question where coaches were asked if they wanted to start in the coach education program the first or the second year, so that the coaches in the control group had agreed to be put “on hold” for 2019. Sixty-six of the coaches (61.7%) applied to start in the program at year one and were thus assigned to the experimental group, whereas 41 (38.3%) applied to start at year two and were assigned to the control group. A pre-test design was applied through an online questionnaire. The coach education program was then carried out for the coaches in the experimental group for the next year. The current study is based on data from the first year of the program.

**The coach education program—formal mentoring of coaches.** The coach education program in the current study was arranged as individual mentoring of coaches, led by educated mentors with substantial experience in elite sports. The aim was to achieve learning among the coaches by including supervised field experiences and facilitate reflection based upon them by experienced coaches who served as mentors. Twenty-six coaches with substantial experience with different sports were recruited to serve as mentors for the coaches who were assigned to the experiment group. The main criteria for selecting mentors for the program was an evaluation done by the NOSC of their experience from coaching at elite level and their experience and competence as mentors. Accordingly, all mentors had to fulfil a mentoring education program carried out by the Department of Education and Life-Long Learning at the Norwegian University of Science and Technology to ensure high quality in the mentor–coach relationship. The program offered 7.5 university points. The course was designed to give the mentors theoretical, practical and research-based knowledge about mentoring based on a coach-centered approach [37]. The program was aimed at improving the communication skills of the mentors, especially their attending skills such as using open-ended questions, stimulating reflections based on the coaches’ own experiences, and listening skills. The goal for the mentors was to develop their ability to guide developmental processes, by facilitating growth and progress based on the coaches’ autonomy and own competence. The educational program for the mentors was conducted in parallel with their mentoring process of coaches. The educational program was carried out by 4 education gatherings, each lasting for two full days, and individual lessons that the mentors had to complete after each gathering. The education program was completed with a final written exam and the aim was to help and support the coaches to develop effective relationships with their athletes.

The group of mentors were divided into 9 groups from 9 different regions in Norway, with 2–5 mentors in each group; each group had the responsibility for 5 to 10 coaches in each region. Each mentor had the responsibility for 1 to 4 coaches in the program and helped and supported the coaches individually. The mentors had 4 team gatherings led by a superior mentor together with their coaches, where the sharing of experience and competence between them was a primary focus, based on person-centered values [37]. The mentors were instructed to execute at least 10 individual consultations with their coaches, based on the principles in their own mentoring education, and participate in observations of them in training and competition situations. Thus, the coach education program was based on athlete-centered values, with a focus on the social process of coaching, such as how to communicate to empower the athlete and use the athletes’ own experiences in the development process. The mentor program in the current study was therefore defined as a formal mentoring program, since it was sanctioned, managed and structured by an organization [35].

**Instruments.** To investigate if the coach program was effective, the current study included variables that measured the coach–athlete relationship and coach performance. The measurements were based on previously developed scales proven to hold satisfactory validity and reliability. The questionnaires included measures for the assessment of the working alliance between the coach and their athletes and perceived coach performance. All measurements were used in Norwegian. The measurements are described below in more detail.

**The Coach–Athlete Working Alliance Inventory (CAWAI).** Coaches’ perception of the coach–athlete working alliance was measured using the working alliance inventory—short version [52,53]. The form has been translated to Norwegian, adjusted and validated to the sports context, e.g., words like therapist, therapy and client were changed to coach, athlete and training [47]. The adjusted form, the Coach–Athlete Working Alliance Inventory (CAWAI), consists of three subscales measuring the different components of the working alliance: agreement concerning goals (CAWAI-goal), agreement on the tasks chosen to achieve the defined goals (CAWAI-task), and the personal bond between the coach and the athlete who receives help (CAWAI-bond). Each subscale has 4 corresponding items relating to goal, e.g., “The coach and athlete are working on mutually agreed upon goals”, task, e.g., “There is agreement about the steps taken to help improve the athlete’s situation”, and bond, e.g., “There is mutual trust between the coach and athlete”. The coaches were asked to respond on a 7-point scale ranging from 1 (never) to 7 (always), indicating to what degree the statement applied to them and their coach–athlete relationships. The Cronbach’s alphas for each subscale were 0.55/0.57 (goal), 0.71/0.75 (task), 0.76/0.69 (bond), and for the complete measurement (CAWAI-sum) 0.84/0.83, at the pre-test and post-test, respectively.

**Perceived Coach Performance (PCP).** An adjusted version of the individual performance from the Athlete Satisfaction Questionnaire was used to measure coaches’ perceived satisfaction with their own performance as coaches in their sports [54]. This subscale seeks to measure the perceived satisfaction with progress in own task performance. Task performance includes a perception of absolute performance, improvements in performance and goal achievement; for example, “I am satisfied with the degree to which I have reached my performance goals during the season”. The coaches were asked to consider 4 items and how satisfied they were with their own progress as coaches in their sports during the last year on a 7-point scale ranging from 1 (not at all satisfied) to 7 (extremely satisfied). The Cronbach’s alphas for the scale were 0.86/0.89 at the pre-test and post-test, respectively.

### Data Analysis

Composite scores for each of the included questionnaires and their respective subscales were calculated according to their relevant scoring manuals. Then, data were analyzed for 107 coaches at the pre-test and 94 coaches at the post-test. Descriptive statistics for the CAWAI-goal, CAWAI-task, CAWAI-bond, CAWAI-sum and PCP, such as statistical means, standard deviations (SD), maximum and minimum values were analyzed as well as a Pearson correlation analysis of the investigating variables. To test the hypothesis in the current study, descriptive statistics including statistical means and standard deviations measuring the investigated variables were carried out for the experimental group and control group, respectively, at each testing time-point. Additionally, paired samples *t*-tests were applied to test for improvements in the investigating variables between pre-test and post-test, in each of the two groups, respectively. To investigate whether the coaches in the experimental group, after receiving help and support from their mentors, had significantly improved the investigating variables, compared to coaches in the control group, a series of five separate hierarchical regression analyses were conducted for each of the dependent variables. The variables at the post-test were included as dependent variables in each of the five hierarchical regression analyses in the respective order: CAWAI-bond, CAWAI-goal, CAWAI-task, CAWAI-sum and PCP. The independent variables entered in the first step were the controlling variables sex and age, and the pre-scores of the investigating variables in each of the five different regression analyses. These variables were entered simultaneously as covariates to rule out their potential confounding effects. In the second step, the group variable was entered as a dichotomized variable. Significance levels were set to *p* < 0.05 for all statistical analyses and all analyses were performed using IBM SPSS (version 25).

## 4. Results

Descriptive statistics of the coaches’ scores and correlations on the coach–athlete working alliance variables CAWAI-bond, CAWAI-goal, CAWAI-task, CAWAI-sum and perceived coach performance (PCP) from the pre-test and post-test are presented in Table 1.

The results from the Pearson correlation analysis shown in Table 1 indicate that there were significant high positive intercorrelations between the different dimensions of the coach–athlete working alliance and the total scale (CAWAI-bond, CAWAI-goal, CAWAI-task, CAWAI-sum), both on the pre-test and the post-test. The correlation analysis also shows that there are significant moderate correlations between all the CAWAI dimensions and the PCP, both at the pre-test and post-test, except from the non-significant low correlation between CAWAI-goal and PCP on the post-test. The Cronbach’s alpha values of the scales are acceptable and good both on the pre-test and post-test, except for a poor Cronbach’s alpha on the CAWAI-goal dimension both at pre- and post-test. Thus, analyses that include the CAWAI-goal dimension should be treated with caution.

To test the hypotheses in the current study, a paired samples *t*-test in was conducted in the experiment group (Table 2) and the control group (Table 3). Means, standard deviations and *p*-values for the paired samples *t*-tests for the outcome variables CAWAI-bond, CAWAI-goal, CAWAI-task, CAWAI-sum and PCP at pre-test and post-test are given in Table 2 and Table 3.

Overall, the paired sample *t*-test showed that there was a significant difference in scores for CAWAI-bond at the pre-test (M = 22.86, SD = 2.48) and CAWAI-bond at the post test (M = 23.54, SD = 2.26) conditions; *t*(56)= −2.35, *p* = 0.022, CAWAI-goal at the pre-test (M = 20.54, SD = 3.08) and CAWAI-goal at the post test (M = 21.46, SD = 3.00) conditions; *t*(56)= −2.09, *p* = 0.041, and CAWAI-sum at the pre-test (M = 65.25, SD = 6.80) and CAWAI-sum at the post test (M = 67.35, SD = 6.73) conditions; *t*(56)= −2.61, *p* = 0.012 (Table 2).

The paired sample *t*-test in the control group showed that there were no significant differences in the scores at the pre-test and post-test for the investigated variables as seen in Table 3.

To test the significant differences that were found in the experiment group based on the paired samples *t*-test, and if the significant effects are effects from the coach-education program, multiple hierarchical regression analyses were conducted for each of the dependent variables (post-test scores for CAWAI-bond, CAWAI-goal, CAWAI-task, CAWAI-sum and PCP). The group variable was entered as a dichotomized variable to test possible group effects. The post-test scores of CAWAI-bond, CAWAI-goal, CAWAI-task, CAWAI-sum and PCP variables were then entered as the dependent variables in five different regression analyses. Sex, age, the pre-scores of the depended variables, and the group variable were entered as independent variables in the regression analyses. The results are presented in Table 4.

The results of the regression analyses indicated that the predictor variables explained 44% of the variance (R^2^ = 0.44, F(4,93) = 19.10, *p* < 0.001) in CAWAI-bond at the post-test, 11% of the variance (R^2^ = 0.11, F(4,93) = 3.90, *p* < 0.01) in CAWAI-goal at the post-test, 30% of the variance (R^2^ = 0.30, F(4,93) = 11.05, *p* < 0.001) in CAWAI-task at the post-test, 33% of the variance (R^2^ = 0.33, F(4,93) = 12.640, *p* < 0.001) in CAWAI-sum at the post-test, and 16% of the variance (R^2^ = 0.16, F(4,93) = 5.51, *p* < 0.01) in PCP at the post-test. It was found that the group variable significantly predicted CAWAI-bond at the post-test (β = −0.24, *p* < 0.01) and that the pre-test score was the largest contributor of the explained variance followed by the group variable. It was found that the group variable significantly predicted PCP at the post-test (β= −0.30, *p* < 0.01) and that the group variable was the largest contributor of the explained variance followed by the pre-test score. As expected, the pre-test scores uniquely predicted the post-test scores (dependent variables), and neither sex nor age had any significant associations with the dependent variables that were entered in the regression analyses.

## 5. Discussion

The current study investigates the effects of a one-year coach education based on a formal mentoring program on coaches’ perceptions of the coach–athlete working alliance through the dimensions bond, goal and task (CAWAI-bond, CAWAI-goal, CAWAI-task, CAWAI-sum), and coaches’ perceptions of their performances as coaches (PCP). The strength of the current study is the experimental design lasting for 1 year with a control group. The hypotheses in the current study predicted that the coach education program would significantly affect the CAWAI-bond, CAWAI-goal, CAWAI-task, CAWAI-sum and PCP variables positively. The main findings from the conducted analyses were: (1) significant positive effects from the experiment on coaches’ perceptions of the CAWAI-bond dimension, and (2) significant positive effects on coaches’ perceptions of the PCP. Thus, the hypotheses in the current study were only partly confirmed.

### 5.1. The Effect on the Coach–Athlete Working Alliance Bond Dimension

The paired samples *t*-tests indicated that there were significant positive differences in the scores at the pre-test and post-test for the CAWAI-bond, CAWAI-goal, and CAWAI-sum variables in the experiment group, whereas no significant differences in the scores at the pre-test and post-test were found in the control group. These findings indicate positive effects from the coach education program. Further analyses were conducted to investigate possible effects from the experiment and control groups, by controlling for potential differences between the two groups at pre-test by entering the group variable as an independent variable in the regression analyses. The results from the regression analyses indicated that there were significant effects from the group variable on CAWAI-bond. Thus, the results from the paired samples *t*-tests and the regression analyses give reason to believe that the coach education program improved the coaches’ perceived relational bond between themselves and their athletes. In this section, the authors will discuss potential explanation for the findings related to the CAWAI dimensions.

Interestingly, the coach-centered approach in the current study focuses on the relational process of coaching, such as the improvement of the coaches’ communication skills, especially their attending skills such as their ability to ask open-ended questions and their listening skills. The ability to ask open-ended questions and actively use listening skills is essential to stimulate athletes’ feelings of being heard and understood by their coaches. The results therefore give reason to believe that it might be the coaches’ improved abilities to listen and ask questions that is key to establishing stronger bonds in the coach–athlete working alliances. Accordingly, it is also found to be key to motivation when the athletes’ experiences, strategies and competences are focused on the learning processes, such as in coach- and athlete-centered learning [42]. Research that investigates effects from coach–client working alliances underscore the importance of an active client in the coaching process, whereas empowerment of the client is key to coaching success [55]. It is well documented that it is essential that coaches establish a close and trustful relationship with their athletes because of the importance to understand how their athletes think, feel and act, and use this information to generate learner-facilitated goals and strategies [1]. Thus, the bond in the coach–athlete working alliances is the glue in the relationship.

A critique that is used about coach education programs is generally directed toward the promotion of athletic achievement where the dominant focus normally is on achieving performance enhancements for the coaches’ athletes [15,56]. Thus, the ability to develop clear goals and effective strategies to reach these goals has in general received attention in coach education programs, whereas the social process of coaching has received scant attention. Since recent research claims that the relational process of coaching is key to influence positive outcomes of the coach–athlete relationship [57,58], a positive significant change in the coaches’ perception of the bond dimension is therefore worth noting. Interestingly, the bond dimension has the highest score among the three dimensions of the alliance at the pre-test, both in the experiment and control group. Thus, the coaches’ abilities to establish strong bonds with their athletes is the strongest skill among the coaches in the current study to begin with. Therefore, this skill might be the most difficult one to influence. Interestingly, the social process of coaching has also been focused on in Nordic countries in recent years, which might explain the highest score on the bond dimension [43,59].

The current result might also indicate that the mentoring process based on a coach-centered approach, where the mentors were taught to use communication skills such as open-ended questioning and active listening, was also adapted implicitly by the coaches since the mentors might have served as role models for the coaches. Research also claims that learning among coaches is most effective when it is interactive, collaborative and located in practice [60]. Thus, the coaches might have experienced the effect of such an approach from their mentors in their own motivation and learning, and because of this self-experience they adopted it. Interestingly, a recent study claims that despite the fact that coach developers acknowledge “learner centered principles”, they find it challenging to implement them in practice [61]. However, the possible association between experiencing an effective learning approach and being motivated to implement it based on self-experience has to be further investigated in future studies.

### 5.2. The Effect on the Coaches’ Perceived Coach Performance

The paired samples *t*-tests that were conducted in the current study indicated no significant differences in the scores at the pre-test and post-test for the PCP variable, neither in the experiment nor the control group. These findings indicate that there were no effects from the coach education program. However, the results from the regression analyses indicated that there were significant effects from the group variable on the PCP variable when potential differences between the two groups at the pre-test were controlled for. In this section, the authors will discuss potential explanation for the findings related to the PCP variable. Thus, the current results indicated that the coaches believe that their coach performances were significantly improved after a year with coach education based on formal mentoring. Thus, the second hypothesis that predicted that the coach education program would affect the coaches’ perceived coach performances positively, was confirmed. Interestingly, the highest correlation in the current study is between coaches’ perceived coach performance and the coach–athlete working alliance dimensions bond and task. The association between perceived coach performance and the goal dimension was not significant at the post-test. A possible explanation of the current result might be that the coaches believe that establishing a strong bonding with their athletes is an essential task for the coaches. The PCP scale seeks to measure coaches’ perceived satisfaction with their task performance development, which includes a perception of absolute performance, improvements in performance and goal achievements. Since coaches’ perceived coach performance has improved during the intervention, coaches’ performance goals might have been related to bonding tasks, and thus they believe that their coach performances have improved. However, the results of associations between perceived coach performance and the different dimensions of the coach–athlete working alliance are inconclusive. It would be a natural consequence that coaches who experience that agreed upon tasks help their athletes to achieve their goals, also believe that the bond between themselves and their athletes is strong. This is also reflected in the correlation coefficients between bond and task, which are large at both the pre- and post-test [62,63].

### 5.3. No Effects on the Coach–Athlete Working Alliance Goal and Task Dimension

The current study did not find any effects of group on the goal and task dimension and the coach–athlete working alliance construct predicts that it is essential to define goals and effective tasks that help athletes toward goal attainment [64]. To become competitive, athletes need to develop their key sport-specific capacities [65] and a recent study showed that the task dimension had the strongest association with athletes’ perceived performance [47]. Athletes’ beliefs in their capabilities to complete task-specific demands are found to be the most predictive variable of performance in several studies [64,66]. Interestingly, while client-initiated agreements on goals and tasks are found to be positively related to coaching success, research from coach–client dyads claims that bonding behavior did not influence coaching successes [55]. Therefore, the current result gives reason to claim that the coach education program in the current study did not fully fulfil its intention.

A potential explanation might be that improving coaches’ abilities to establish stronger bonds through developing their attending skills such as listening and asking open questions [41,67], are valued as central coaching tasks among the coaches in the current study. However, the ability to ask powerful questions that reveal both conscious and unconscious information that is highly relevant for sport-specific improvements, and at the same time do not hamper the bond in the relationship, is another advanced skill within communication. This skill is defined as influencing skills [67]. Coaches often need to influence the athletes’ motivation and behaviors in order to help their athletes to achieve changes. Influencing skills seek to increase awareness about highly relevant tasks and actions that need to be improved to achieve such changes. Typical influencing skills are techniques such as interpretation, confrontation, direct advice, recommendations and instructions [68]. Thus, agreement on goals and tasks might be a more difficult and demanding endeavor than creating stronger bonds since it requires systematic knowledge about sport-specific demands and learning. Powerful questioning is therefore claimed to be one important influencing skill in the coaching process [46], and the current results might indicate that the coaches did not significantly improve this skill.

This should be an important issue in future coach education. Thus, to achieve growth and development, coaches need to have the coaching skills necessary to produce changes in the task domains that are sport specific. Other helpful relationship education programs, such as for psychologists and therapists, must undergo hundreds of hours with communication training in their education to develop their skills in communication. Elite athletes are normally not struggling with mental illnesses, but they operate in a highly stressful environment because of the continues demands for improvements and being competitive. The question is if coach education programs should pay more attention to the social process of coaching by focusing on how coaches implement their communication with their athletes, both their attending and influencing skills?

### 5.4. Limitations

While the current study has interesting results, it also has some limitations. An important limitation in the current study is that the data are based on self-reports from coaches. The authors cannot know if their athletes evaluate their coaches in a similar manner. Importantly, all relationships are dyads between two or more participants, and information about the athletes’ perspectives would have strengthened the study. A recent study also shows that coach education might affect coaches’ knowledge, but questions if coaches manage to bridge the knowledge–practice gap with their athletes [69]. However, research investigating the effectiveness of coach education is often based on qualitative interviews of coaches, and their critique of formal coach education is clear [14,60]. Thus, findings based on self-reporting should be noted. Another limitation that should be considered is the mentoring process of the coaches. Even though the mentors were experienced in the field of sport coaching and undertook a formal University program, no data from the mentors are provided in the current study.

## 6. Conclusions

The results in the current study indicate that the coaches believe that their performances as coaches have improved. Importantly, it seems like the coaches’ perceived improvements are related to the social bonding process between themselves and their athletes, which Cushion [70] suggests needs to be emphasized in the field of coaching in sport. Thus, the current topic is understudied within sport science and therefore deserves scientific attention [35]; studies with strong designs such as the experimental design with a control group that was conducted in the current study are particularly needed.

## Figures and Tables

**Table 1 sports-09-00003-t001:** Pearson correlation coefficients between the investigated variables and descriptive statistics based on the pre- and post-test for the coaches in the current study (*n* = 107/*n* = 94).

Variables		1	2	3	4	5
1	CAWAI-bond	-				
2	CAWAI-goal	0.55 */0.49 *	-			
3	CAWAI-task	0.64 */0.61 *	0.65 */0.64 *	-		
4	CAWAI-sum	0.84 */0.80 *	0.87 */0.86 *	0.87 */0.87 *	-	
5	PCP	0.43 */0.44 *	0.36*/0.17	0.41*/0.36 *	0.46 */0.37 *	-
Mean		22.72/23.06	20.88/21.49	22.07/22.21	65.67/66.77	19.52/19.85
SD		2.57/2.31	3.00/2.87	2.40/2.33	6.86/6.35	3.38/3.88
Maximum		28/28	28/28	28/28	82/84	27/28
Minimum		17/17	14/14	16/15	51/54	8/4
Cronbach’s alpha		0.76/0.69	0.55/0.57	0.71/0.75	0.84/0.83	0.86/0.89

Note: CAWAI = Coach–Athlete Working Alliance Inventory, PCP = perceived coach performance, SD = standard deviation, * *p <* 0.01

**Table 2 sports-09-00003-t002:** Descriptive statistics (mean and SD) and *p*-values from the paired sample *t*-test for the experiment group at pre-test and post-test (*n* = 57).

Variable	Pre-Test		Post-Test		
	Mean	SD	Mean	SD	*p*
CAWAI-bond	22.86	2.48	23.54	2.26	0.022 *
CAWAI-goal	20.54	3.08	21.46	3.00	0.041 *
CAWAI-task	21.84	2.34	22.35	2.45	0.066
CAWAI-sum	65.25	6.80	67.35	6.73	0.012 *
PCP	19.69	3.24	20.78	3.77	0.072

Note: CAWAI = Coach–Athlete Working Alliance Inventory, PCP = perceived coach performance, * *p* < 0.05.

**Table 3 sports-09-00003-t003:** Descriptive statistics (mean and SD) and *p*-values from the paired sample *t*-test for the control group at pre-test and post-test (*n* = 37).

Variable	Pre-Test		Post-Test		
	Mean	SD	Mean	SD	*p*
CAWAI-bond	22.41	2.76	22.30	2.76	0.731
CAWAI-goal	20.89	2.76	22.30	2.60	0.255
CAWAI-task	22.30	2.60	21.95	2.17	0.404
CAWAI-Sum	65.60	7.10	65.76	5.73	0.870
PCP	18.67	3.54	18.38	3.69	0.620

Note: CAWAI = Coach–Athlete Working Alliance Inventory, PCP = perceived coach performance, * *p* < 0.05.

**Table 4 sports-09-00003-t004:** Summary of linear regression analysis for variables predicting the dependent variables (*n* = 94).

Depended Variable	Independent Variables	B	*t*	*p*	R^2^
CAWAI- bond- post	Sex	−0.055	−0.711	0.479	
	Age	0.106	1.309	0.194	
	CAWAI- bond- pre	0.603	7.664	0.000 *	
	Group	−0.241	−2.967	0.004 **	0.44
CAWAI- goal- post	Sex	−0.075	−0.764	0.447	
	Age	−0.080	−0.780	0.437	
	CAWAI- goal- pre	0.371	3.737	0.000 *	
	Group	0.010	0.094	0.926	0.11
CAWAI- task- post	Sex	−0.116	−1.342	0.183	
	Age	−0.018	−0.200	0.842	
	CAWAI- task- pre	0.561	6.447	0.000 *	
	Group	−0.132	−1.460	0.148	0.30
CAWAI- sum- post	Sex	−0.087	−1.027	0.307	
	Age	−0.012	−0.135	0.893	
	CAWAI- sum- pre	0.577	6.773	0.000 *	
	Group	−0.134	−1.527	0.130	0.33
PCP- post	Sex	0.032	0.343	0.732	
	Age	0.148	1.478	0.143	
	PCP- pre	0.266	2.725	0.008 **	
	Group	−0.301	−3.013	0.003 **	0.16

Note: CAWAI = Coach–Athlete Working Alliance Inventory, PCP = perceived coach performance, * *p* < 0.001, ** *p* <0.01.

## Data Availability

Data sharing is not applicable to this article.

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
