# Peer review of "Investigating Possible Effects from a One-Year Coach-Education Program"

_sports, 2020, doi:10.3390/sports9010003_

Round 1

Reviewer 1 Report

Introduction

The argument line must be concise ending with a concrete aim. What do you want to analyse?

Provide a valid study hypothesis

Methods

Describe the sample analyzed

The procedure to reach the study aims is not clear

Explain the analysis conducted to reach the study aims

Results

Explain tables, they must be interpreted by themselves regardless of the main text

Discussion

Focus on results, discuss results according to the aims

Conclusion

Responding aims concisely

Author Response

The authors would like to thank the reviewer for constructive and important feedback to improve the manuscript. All suggested revisions are accepted in the revised manuscript and the authors hope that the suggested revisions meet the suggested claims for changes.

  1. The introduction is shorter, and it is ending with the main arguments why scientific studies that investigate possible effects from coach education is important, as suggested. A shorter introduction with the main arguments is organized as a paragraph of its own, followed by a theoretical paragraph.
  2. The aim of the study is defined more concrete at the end of the theoretical review, and the theoretical arguments are ending with a concrete aim, as suggested.
  3. The sample that is analyzed is in the revised manuscript described early in the method section as suggested.
  4. The procedure to reach the study aims is improved and the analysis that is used is explained in more detail, as suggested.
  5. The tables are explained in more detail as suggested.
  6. The discussion part is rewritten and focuses more clearly on the results according to the hypothesis and aim for the study, as suggested.
  7. The conclusion is revised and is more concisely.

Reviewer 2 Report

Dear Authors,

It was pleasure to read this paper. Article discuss relevant and interesting topic in both theoretical and scientific level.

Although the study has much potential, there are also several limitations that need to be addressed before the manuscript could be recommended for publication.

The abstract in general is well written.

I suggest shortening the introduction of the article, leaving only the relevance of the topic; research problem and consideration and the level of its investigation and, of course, aim, objectives, hypothesis.

I suggest moving the other information provided in the introductory part to a new part - theoretical part of the topic or similar. This section could provide a deeper theoretical analysis to answer the question of why the authors chose such variable for analysis as “The coach-athlete working alliance”.  The theoretical part would also be strengthened by analyzing variable such as coaches’ perceptions on their own performance.

The Methods part needs small corrections. The criteria for selecting a mentor are not entirely clear. What is meant by the statement “educated mentor”. Whether the primary criterion for selecting a mentor was just “long experience”. Whether his education and competence development was not determined during the selection (if no, why?).

Part of the discussion discusses a particular issue in coaching education, but it is not clear whether it is about formal or non-formal education.

Particular attention should be paid to references - sources should be provided as required.

Author Response

The authors would like to thank the reviewer for constructive and important feedback to improve the manuscript. All suggested revisions are accepted in the revised manuscript and the authors hope that the suggested revisions meet the suggested claims for changes.

  1. The introduction is shortened, by making a shorter paragraph initially, and making a paragraph for a theoretical review that is made shorter, as suggested.
  2. The criteria for selecting mentors for the program is clarified, as suggested.
  3. The particular issue in coaching education is defines, as suggested, and the discussion part is rewritten to make the part more structured.
  4. The references are revised as suggested.

Reviewer 3 Report

The paper is an experimental study on the effect of a coaches mentoring program on the improvement of coaching. The overall results were that the mentoring was effective in improving the coach-athlete relationship (bond) while was not effective in improving the performance (goal and task dimensions).

The paper is interesting because the coach training is poorly investigated in scientific literature, and it evidences how it can be improved. Strengths are a correct methodological approach and sound methods, while weakness are that the results are probably less than expected, because overall in the sport the results is the most important outcome, and the mentoring programs does not improved the athletic results. This can be due in part to the fact that the involved coaches operates yet at an high level (e.g. Norway national teams).

In my opinion, there are not further minor issues to be corrected and paper can be published, albeit the overall quality is not excellent but average/good because of the minor findings. The paper can be of high interest for those who design training programs for coaches and it study a often underrated topic.

Author Response

The authors would like to thank the reviewer for constructive and important feedback on the manuscript. The authors have revised the manuscript to make it clearer regarding aim, hypotheses and results, and English language is revised as suggested.

Reviewer 4 Report

This paper may reach a publishable level only with substantial revisions.

My comments/questions:

The topic is interesting to readers.

The authors present experimental study data in 1, 2, 3, and 4 tables but discuss them insufficiently. I am unsure: did experiment data confirm the hypothesis that "the key responsibility for a coach in high-performance sport is to ensure athletes’ growth and learning and increasing their competitiveness".

The discussion is not sufficient and does not based on the experimental data and does not compare them with the data of other authors.

The findings haven't been done based on research data.

In conclusion, I would like to point out that the manuscript requires a description of the results of the study, drawing the readers' attention to the most important data of the study.

The discussion and conclusions should be directly linked to the research data. It is necessary to base the conclusions statements made by the authors on the research data.

The conclusion of this review is that the manuscript should be improved and research data have to be described more clearly and precisely. The conclusions should provide links between the main data of this study and the discussion text.

Author Response

The authors would like to thank the reviewer for constructive and important feedback to improve the manuscript. All suggested revisions are accepted in the revised manuscript and the authors hope that the suggested revisions meet the suggested claims for changes.

  1. The authors have explained the tables in more detail, and discussed them more sufficiently in the revised manuscript, as suggested.
  2. The revised version has defined the hypotheses more clearly, the analyses conducted more clearly, and discusses the results from the analyses more structured and according to the hypotheses and results, as suggested.
  3. The conclusion is more concisely and summarizes the main findings, as suggested, and the conclusion is based on the research data.
  4. The revised manuscript is describing the aim, hypothesis, and relates the research data and results more clearly to the hypothesis, as suggested. The discussion more clearly relates hypothesis and research results together, as suggested.

Round 2

Reviewer 1 Report

Authord made all the corrections required

Author Response

The authors would like to thank the reviewer for accepting the revisions that were suggested in the second manuscript.

Reviewer 2 Report

Dear authors. Thank you for your efforts in improving the article.

Author Response

The authors would like to thank the reviewer for accepting the suggested revisions in the revised manuscript. 

In the resubmitted manuscript the results are more clearly addressed in the result section of the manuscript, 

  • page 6, after Table 1, the first paragraph is completely rewritten.
  • page 6, second paragraph, the analysis is more clearly described.
  • page 7, after Table 2, the first paragraph is completely rewritten.
  • page 7, after Table 3, the first and second paragraph is completely rewritten.
  • page 8, after Table 4, the first paragraph is completely rewritten and statistics results are more clearly addressed.

The conclusion part of the manuscript is more clearly addressing the hypotheses in the current study and relate them more clearly to the results in the result section. 

  • page 9, first paragraph in the conclusion part, small changes are done.
  • The first heading is more clearly addressing hypothesis 1 in the study, and the 13 first lines are completely rewritten to make the connections more clear.
  • page 10, the first heading is revised and is more clearly addressed to hypothesis 2, and the first 9 lines in the paragraph are rewritten to address the results more clearly.
  • Page 10, the third heading is also revised, and is more clearly addressed to the non significant findings in the study.
  • page 11, the conclusion is revised and is more clearly addressing the results in the current study.

Best regards

The authors 

Reviewer 4 Report

Thanks to the authors for the corrections. However, their efforts are insufficient.
Therefore, my comments remained largely unchanged:
1. The authors presented experimental study data in table 1, table 2, table 3 and table 4  but didn't discuss them sufficiently in section nr 4. Results.
2. The findings haven't been done based on research data.
The conclusions of this manuscript should be detailed. The scope of the conclusions is insufficient.

Author Response

(The authors gave the same response as above.)

Round 3

Reviewer 4 Report

Thank for authors, they did corrections according to my comments.

Sincerely.